# Climate policy support as a tool to control others' (but not own) environmental behavior?

**Charlotte A. Kukowski**[1]*, **Katharina Bernecker**[1], **Leoni von der Heyde**[2], **Margarete Boos**[2], **Veronika Brandstätter**[1]

**1** Department of Psychology, University of Zurich, Zurich, Switzerland, **2** Department of Psychology, Georg-August-University of Göttingen, Göttingen, Germany

* c.kukowski@psychologie.uzh.ch

**Data Availability Statement:** All data, materials, and code are available from the Open Science Framework at DOI 10.17605/OSF.IO/87325 (https://tinyurl.com/3jsk87re).

## Abstract

Drastic reductions in greenhouse gas emissions are necessary to successfully mitigate climate change. Individual environmental behavior is central to this change. Given that environmental behavior necessitates 1) effortful individual self-control and 2) cooperation by others, public policy may constitute an attractive instrument for regulating one's own as well as others' environmental behavior. Framing climate change mitigation as a cooperative self-control problem, we explore the incremental predictive power of self-control and beliefs surrounding others' cooperation beyond established predictors of policy support in study 1 using machine-learning ($N$ = 610). In study 2, we systematically test and confirm the effects of self-control and beliefs surrounding others' cooperation ($N$ = 270). Both studies showed that personal importance of climate change mitigation and perceived insufficiency of others' environmental behavior predict policy support, while there was no strong evidence for a negative association between own-self control success and policy support. These results emerge beyond the effects of established predictors, such as environmental attitudes and beliefs, risk perception (study 1), and social norms (study 2). Results are discussed in terms of leveraging policy as a behavioral enactment constraint to control others' but not own environmental behavior.

## 1. Introduction

The climate crisis is imminent. Wildfires, tornadoes, heatwaves, droughts, flooding, and other extreme weather events are becoming increasingly common, resulting in an unprecedented threat to entire regions and the people who inhabit them [1]. Several of the world's large land and ocean ecosystems, such as the Amazon rainforest and the West Antarctic ice shield, are approaching tipping points of irreversible damage. A cascade of such tipping events would make large parts of the planet uninhabitable [2].

On the road to a more sustainable future, substantial shifts in many areas of life will become necessary, ranging from the food we eat to the energy that powers our homes, from the modes of travel we choose to our consumption patterns. As pointed out by Nielsen [3], the individual

**Funding:** The authors received no specific funding for this work.

**Competing interests:** The authors have declared that no competing interests exist.

citizen is at the heart of this change: Accumulated everyday actions snowball, adding up to a considerable impact of individual-level behaviors on environmental outcomes [4]. As with other collective-action problems like COVID-19 mitigation, where individuals must incur short-term personal costs for long-term collective pay-off [5] alongside potential benefits, engaging in pro-environmental behaviors requires individuals to exert self-control to advance a societal goal. In other words, individuals must overcome conflicting desires and impulses (e.g., for comfort) for the sake of collective goal attainment. However, self-control is effortful, and individuals often do not succeed in implementing pro-environmental behavior despite holding pro-environmental goals [i.e., self-control failure, 3, 6, 7].

Given the difficulty in self-regulating individual environmental behavior, behavioral policy constitutes a promising pathway toward urgently needed large-scale change [8, 9]. By incentivizing or mandating environmentally helpful behaviors and making environmentally harmful behaviors difficult or impossible to enact, behavioral policy steers individual behavior toward the societal climate change mitigation goal. For instance, behavioral policy can increase prices on environmentally harmful behaviors such as air travel and meat consumption, or make them more difficult, e.g., by banning cars from city centers. In doing so, behavioral policy externally regulates environmentally relevant behavior, thereby freeing individuals from self-control conflicts to which they might otherwise succumb [10] and markedly increasing the likelihood of pro-environmental behavior implementation. This necessity of supplementing self-regulated behavior with external intervention to steer individual action is characteristic of large-scale collective-action problems [5]. For instance, in the COVID-19 context, collective goal pursuit, that is, pandemic mitigation, similarly requires individual self-controlled behavior in combination with behavioral policy to steer individual efforts (see [11]).

Given the usefulness of behavioral policy in combating climate change through large-scale alignment of individual behavior with the societal mitigation goal, it is important to understand the factors that shape public support for such policy (e.g., [12, 13]). That is, under which circumstances do individuals support (vs. oppose) government regulation of individual environmentally relevant behavior? Our interest in predicting policy support is threefold. First, support for behavioral policy is theoretically interesting in that individuals agree to external regulation of their own and others' behavior that would otherwise be self-regulated. Second, we assume that policy regarding individual pro-environmental behavior facilitates the successful implementation of such behavior. Conversely, low policy support may lead to insufficient compliance, rendering policy less effective. Third, the successful implementation of behavioral climate policy requires the support of the public. Indeed, the historical record shows [14] that low public support for government measures can lead to social division, protesting, and political unrest [15]. Especially in democracies with direct participation elements (e.g., referendums), support is needed for citizens to vote for or sign petitions aiming to implement climate policy. In turn, democratically backed policies can have positive downstream effects on compliance and acceptance of further political instruments, as recently observed in declining numbers of protests following the majority acceptance of the Swiss COVID referendum, possibly due to perceptions of dominant public opinion via one's social environment, as proposed in spiral of silence theory [16].

## 1.1. Climate change mitigation as a cooperative self-control problem

From a social psychological perspective, climate change mitigation constitutes a cooperative problem [8, 17] that requires individual self-control (see [3] for a review). Individuals must exert self-regulatory effort [10, 18], incurring personal costs (e.g., sacrificing comfort) for collective payoff [19], for a review, see [15]. Such self-control effort and success become more

likely when the personal goal is valued, that is, of importance to the individual (value choice theory, [20]). Notably, pro-environmental self-control is embedded into a societal context and only becomes effective when compounded: Contrary to an individual-level goal, such as health or academic achievement (e.g., [21]), self-control in pro-environmental behavior primarily serves the collective benefit, individual benefits only materializing if collective behavior is sufficient to bring about the desired outcome. Therefore, in contrast with other self-control problems, others' behavior is relevant to the attainment of the collective goal state (i.e., mitigating climate change). Our conceptualization of pro-environmental behavior as a cooperative self-control problem is consistent with a growing body of literature indicating that cooperative behavior requires individual-level self-control [22]. Previous work has shown that the same factors underlying self-control in long-term personal goals enhance behavior for collective over individual interests [23] and that those people higher in trait-self-control are more likely to cooperate in public goods games [24, 25].

While standard definitions of self-control highlight its role in fostering societally beneficial behavior in addition to its relevance to individual goal-striving [26], the literature has focused on self-control deployed in the service of individual-level goals. Thus, despite its definition as a capacity that helps people to align their interest with that of others, self-control research has largely ignored the broader societal context into which it is embedded (for an exception, see [27]). The present work aims to close this gap in the literature by exploring in a first study and then confirming in a second hypothesis-testing study whether individual self-control predicts support for climate mitigation policy. Based on integrative self-control theory [10], we include *trait self-control*, that is, domain-independent habitual self-control capacity, *personal goal importance*, that is, the extent to which climate change mitigation is a valued personal goal, and *self-control struggle*, that is, the extent to which individuals have difficulty enacting pro-environmental behavior given opposing impulses and temptations.

## 1.2. Policy support as tool to affect other people's behavior

Extant work examining climate policy support has highlighted either descriptive norms (i.e., what others are doing) or injunctive norms (i.e., other' expectations) regarding environmentally relevant behavior [8, 28–31]. However, it remains unclear how perceptions of the sufficiency and importance of others' environmental behavior, that is, whether others are perceived to be doing *enough* to address climate change, associates with support for policy. Classic research on collective action problems suggests that individuals will contribute to a common resources if others do as well, a phenomenon known as contingent consent [32, 33]. Individuals are therefore motivated to align others' behavior with the collective goal and to punish those who do not cooperate, even at a cost to themselves [34], across cultures [35]. It has additionally been found that groups will vote to restrict individual solutions to collective action problems in favor of collective solutions to bring about cooperation [36]. Such external regulation of individual behavior may be an attractive means of aligning individual collectively relevant behavior, and it has indeed been shown that individuals may see climate policy as a means of externally regulating and restraining environmental behavior for large-scale behavior change in qualitative studies [37, 38]. Recent work on COVID-19 policy substantiates this point, showing that perceiving others as insufficiently compliant with health-protective behaviors is associated with increased support for policy, presumably to regulate such behavior [11]. This aligns with classic work showing that individuals perceive others as more malleable by social influence than themselves [39]. To test the idea that perceptions of the sufficiency and importance of others' cooperation in mitigating climate change may be associated with support for policy that regulates mitigation behavior, we include two indicators. Specifically, we

measure *perceived insufficiency* in others' environmental behavior, i.e., the perception that others are not doing enough to protect the climate, and *concern with cooperation*, i.e., a preoccupation with others' contributions to climate mitigation.

## 1.3. Previously established predictors of policy support

A substantial body of work has connected a range of environmental attitudes to support for behavioral climate policy. Here, we include two such attitudes as control variables, environmental concern and perceived behavioral control, as these have been identified as relatively proximal predictors of pro-environmental behavior and policy support [40]. Environmental concern refers to an evaluation of the seriousness of environmental problems [41] rooted in individuals' value systems (Schultz, 2000, 2001, as cited in [42]) and has been identified as the most immediate antecedent of attitudes towards environmentally relevant policy objects, such as green energy [40].

Further, a range of pro-environmental behaviors has been linked to perceived behavioral control (also: perceived consumer effectiveness), which describes an individual's belief that they can make a meaningful contribution to environmental conservation [43]. Conversely, low perceived behavioral control over a given behavior reduces behavioral intentions, even when attitudes and norms toward the behavior are positive [44]. Indeed, early research has shown that those who are highly concerned about the environment but who perceive a low individual ability to make a difference in environmental conservation are more supportive of government regulation to "force people to protect the environment" [43]. We, therefore, include environmental concern and perceived behavioral control as control variables in study 1.

A host of studies has identified a relatively strong link between the perceived risk of climate change and support for action to address it. Prominent theorizing distinguishes between analytic risk perception, which is informed by probabilistic and logical judgment, and affective risk perception, which is guided by affectively informed images and associations [45]. Both types of risk perception have been linked to policy support. For instance, appraisal of negative climate change consequences for oneself and others predicts greater policy support [46–48], as do discrete risk-related emotions, particularly worry [49]. Building on their emergence among the top five most important predictors of climate policy support in Goldberg and colleague's 2020 article [50], we include both affective (*climate change-related anxiety and distress*) and analytic perceptions of risk (*perceived risk*) as control variables in study 1. We additionally control for three sociodemographic variables, political orientation, gender, and age.

## 1.4. The present research

Framing climate change mitigation as a cooperative self-control problem, we aim to shed light on how individuals' own self-control and their perception of others' cooperation in combating climate change associate with their support for behavioral climate policy. We do so in two parallel studies, separating exploration (study 1) and pre-registered replication and extension of the same effects (study 2) to mirror the structure of the scientific process to increase the robustness of our findings. In study 1, we explore whether self-control and perception of others' cooperation explain variance beyond the eight previously established psychological and sociodemographic predictors outlined above. We use elastic net regression, an exploratory machine learning-based procedure that selects out variables that do not contribute independent predictive validity to the model, to identify the predictors that independently account for the largest proportion of policy support variance. Having established their incremental predictive power in study 1, we replicate the effects of self-control and cooperation variables on policy support in a new sample in pre-registered study 2. We additionally extend these

associations to specific policies based on Goldberg and colleagues' recent paper [50]. This procedure is based on other studies examining large numbers of predictor variables [51, 52]. Further, separating exploration and confirmation is important given that overfitting hypothesis-testing models in combination with flexible data analysis has been identified as a major contributor to the replication crisis in psychology [53–55].

## 2. Study 1

Study 1 aimed to explore whether self-control and cooperation variables would emerge as independent predictors of behavioral climate policy support beyond previously identified variables. Notably, past research has often considered these predictors' effects on policy support in isolation (for an exception, see [50]). This study met its set aim by expanding upon past work by identifying important predictors of policy support from a theoretically and empirically informed pre-selection using a data-driven variable selection approach. We gauge the effect size of the selected predictors using OLS regression analysis.

### 2.1. Method

**2.1.1. Samples and study designs.** We collected data from two independent samples. We control for sample (A or B) in all analyses and find only zero-order, statistically non-significant effects on our outcome. To improve power and given that the study procedure and materials were identical, the main analyses for study 1 were conducted for the pooled sample ($N$ = 610). Sensitivity analyses using G*Power 3.1 indicate that the study has 80% power to detect effect sizes equal to or larger than $f^2$ = .02, which covers all effects of interest in the current study (smallest effect size: $\beta_{\text{self-control struggle}}$ = -.05 = $f^2$ = [.11]. We obtained approval from both universities' ethics boards before commencing data collection (numbers 21.2.7 and 242, respectively).

*2.1.1.1. Sample A.* For sample A, $n$ = 358 German-speaking adults residing in Switzerland completed the survey for this project. Participants were recruited via social media, flyers, online forums, mailing lists, and word-of-mouth. Out of the sample with complete questionnaire data, we excluded $n$ = 6 participants who failed a one-item data quality check, asking whether they had responded truthfully and conscientiously while assuring them that their answers would not affect their compensation. We additionally excluded $n$ = 2 minors. Further data quality checks assessing completion time (minimum 150 seconds without interruptions) indicated no problematic data points. The final sample ($n$ = 350) was 73% female (*M*age = 28.03 years, *SD*age = 11.06 years, range = 18–79 years). Participants enrolled in an undergraduate psychology program (46%) were offered course credit as compensation. Participants could additionally enter a raffle with the chance to win one out of five vouchers to an online retailer, valued at CHF 50 (USD 54) each. Concerning employment, 49% were employed full- or part-time, and an additional 39% were in training (university or technical school), 5% unemployed, 1% retired, 1% homemakers, and 5% none of these. Sample size was set as the maximum possible number to be obtained until the end of the semester (04–06/2020).

*2.1.1.2. Sample B.* For sample B, $n$ = 267 adults residing in Germany completed the questionnaire. After exclusions due to unsolicited participation by minors ($n$ = 2) and data quality checks as described above ($n$ = 5), the final sample ($n$ = 260) consisted of 71% women (*M*age = 27.85 years, *SD*age = 9.67 years, range = 18–78 years). Again, participants enrolled in an undergraduate psychology program (54%) were offered course credit as compensation. Regarding employment, 45% were employed full- or part-time. The remaining half of participants were either in training (34%), unemployed (4%), retired (0.01%), homemakers (0.01%),

or none of these (16%). Participants could also enter a raffle for three 30€ (USD 35) vouchers to online stores of the participants' choice. The sample was recruited via social media, online forums, mailing lists, and word-of-mouth.

**2.1.2. Measures and procedure.** Participants completed a 15-minute online survey on the German survey platform SoSci-Survey. The study initially included an experimental manipulation, which we control for in all analyses, and additional variables to collect data for another research question. These data are therefore used exclusively for exploration, and the materials not included in this study are available on the OSF, alongside the data and materials (https://tinyurl.com/3jsk87re).

*2.1.2.1. Dependent measure*: *General support for behavioral climate policy.* Participants filled in a twelve-item scale ranging from 1 (*do not agree at all*) to 5 (*completely agree*), with five reverse-scored items. The scale was adapted to the environmental context from [11], e.g., "I support government regulation of individual behavior to tackle climate change"; "When it comes to climate change, I would prefer it if citizens were left to regulate their own behavior, without the intervention of legislators" (reversed; see "Descriptive Statistics and Zero-Order Correlations" for reliability information for all scales). Higher scores indicate higher support for behavioral climate policy.

2.1.2.2. Self-control related predictors.

*2.1.2.2.1. Trait self-control.* Participants completed the (German) short version of the Trait Self-Control Scale [56, 57] ranging from 1 (*not at all*) to 5 (*very much*). The scale consists of 13 items, e.g., "I am good at resisting temptations."

*2.1.2.2.2. Personal goal importance.* To assess participants' personal importance of climate change mitigation, we asked them to consider "the goal of doing something against climate change in your everyday life." Participants indicated the extent of their agreement with each of five items on a scale ranging from 1 (*do not agree at all*) to 5 (*completely agree*), e.g., "This goal is important to me" (adapted to the environmental context from [11]).

*2.1.2.2.3. Self-control struggle.* Participants read a short introduction describing pro-environmental behavior. For each of five pro-environmental behaviors (e.g., recycling, flying less), they indicated the extent to which they "struggled to implement these behaviors, for instance, because they are cumbersome or because the alternative is more fun" on a scale ranging from 1 (*not at all*) to 5 (*very much*). The measure was adapted to the environmental context from [58].

2.1.2.3. Beliefs surrounding others' cooperation.

*2.1.2.3.1. Perceived insufficiency.* To measure participants' perception of insufficiency in others' pro-environmental behavior, we administered a three-item scale (e.g., "I think that other people are not doing enough about climate change.") ranging from 1 (*not at all*) to 5 (*very much*) (adapted to the environmental context from [11]). Higher scores indicate higher perceived insufficiency in others' pro-environmental behavior.

*2.1.2.3.2. Concern with cooperation.* To assess the extent to which individuals are concerned with others' contributions to climate change mitigation, we administered a three-item scale (e.g., "I think it is unfair when other people do not behave in an environmentally friendly way."; adapted to the environmental context from [11]). The scale ranged from 1 (*do not agree at all*) to 5 (*completely agree*).

2.1.2.4. Environmental attitudes and beliefs.

*2.1.2.4.1. Environmental concern.* To measure participants' level of environmental concern, we administered a four-item scale with items such as "If things continue on their present course, we will soon experience a major ecological catastrophe" (1 = *do not agree at all*, 5 = *completely agree*; [41]).

***2.1.2.4.2. Perceived behavioral control.*** Participants indicated the extent of their agreement with each of four items on a scale ranging from 1 (*do not agree at all*) to 5 (*completely agree*), e.g., "There is a lot that any one individual can do about the environment" [41].

2.1.2.5. Climate change-related risk perception.

***2.1.2.5.1. Perceived risk for self and close others.*** Participants were asked how likely they believed it was that they themselves or a loved one would suffer negative consequences of climate change in the future (1 = *very unlikely* to 6 = *very likely*). We originally planned to include these two items as separate predictors, as previous research has shown that people appraise risk differently for themselves and close others [59]. However, since bivariate correlations were extremely high ($r_{\text{Sample A}}$ = .86, $r_{\text{Sample B}}$ = .93), we merged the two items and included this variable as a single predictor.

***2.1.2.5.2. Anxiety.*** To measure anxiety concerning climate change, we administered a seven-item scale adapted to the environmental context from [11], e.g., "How preoccupied are you with thoughts about climate change?" (1 = *not at all*, 6 = *very much*).

***2.1.2.5.3. Distress.*** To measure distress about the consequences of climate change, we administered a shortened three-item measure based on [60], e.g., "At times, I feel overwhelmed when thinking about the future impact of climate change," (1 = *strongly disagree*, 6 = *strongly agree*).

2.1.2.6. Sociodemographic variables.

***2.1.2.6.1. Political orientation.*** We measured generalized political orientation using the left-right self-placement scale [61]. Participants were given a short description of the terms "left" and "right" and asked to localize their political views on a scale ranging from 1 (*left*) to 11 (*right*).

***2.1.2.6.2. Gender.*** Participants chose the gender they identified with (1 = *female*, 2 = *male*, 3 = *non-binary*).

***2.1.2.6.3. Age.*** Participants indicated their age in years.

## 2.2 Results

**2.2.1. Analyses.** In a two-step procedure based on [53], we first applied five-fold cross-validated elastic net regression [62] to identify the most important predictors of climate policy support. Elastic net is a variable selection method based on a technique called regularization. Regularization is able to create parsimonious, well-fitted models from large numbers of correlated predictors by pushing coefficient estimates toward zero, reducing overfitting by selecting out redundant variables. As a technique combining Lasso and Ridge regression, elastic net combines Lasso and Ridge penalty terms into one hyperparameter, α, which determines the amount of mixing between the two, and λ, which is the regularization parameter and thus determines the amount of shrinkage toward zero in the model coefficients [62]. Coefficients that are "selected out" of the model by the elastic net algorithm are referred to as regularized coefficients in the regression table.

While OLS regression analysis has difficulty handling multicollinearity, which emerges when including multiple overlapping predictor variables, elastic net is far more useful in identifying each predictor's *independent* contribution to explaining variance in the dependent variable [62]. We followed up elastic net by running multiple regression models to gauge effect sizes of the selected predictors. This approach allowed us to consider all predictors simultaneously and estimate their incremental value in predicting policy support.

**2.2.2. Descriptive statistics and zero-order correlations.** To provide as much detail as possible, we provide separate descriptive statistics and correlations for the two samples. Average general support for climate policy was moderate to high in both samples ($M_A$ = 3.60, $SD_A$

**Table 1. Summary of intercorrelations, internal consistencies, means, and standard deviations for all study variables by sample.**

| | | 1 | 2 | 3 | 4 | 5 | 6 | 7 | 8 | 9 | 10 | 11 | 12 | 13 | 14 | α | M | SD |
|---|---|---|---|---|---|---|---|---|---|---|---|---|---|---|---|---|---|---|
| 1 | General support | | -.10 | **.45** | **-.18** | **.46** | **.37** | **.37** | .14 | **.37** | **.33** | **.33** | **-.35** | .01 | **-.11** | .89 | 3.60 | .71 |
| 2 | Trait self-control | -.03 | | .12 | **-.21** | -.05 | -.04 | .00 | .01 | -.02 | .06 | -.06 | .10 | **.26** | -.03 | .83 | 3.24 | .63 |
| 3 | Goal importance | **.35** | .09 | | **-.40** | **.32** | **.42** | **.53** | **.38** | **.50** | **.68** | **.55** | **-.27** | .10 | **-.14** | .90 | 3.37 | .90 |
| 4 | Self-control struggle | **-.23** | **-.13** | **-.50** | | -.11 | -.15 | **-.22** | **-.28** | -.15 | **-.25** | -.14 | **.20** | -.12 | .08 | .46 | 2.60 | .67 |
| 5 | Perceived insufficiency | **.40** | -.03 | **.30** | -.14 | | **.38** | **.34** | .13 | **.22** | **.27** | **.28** | **-.20** | -.13 | -.07 | .66 | 3.55 | .72 |
| 6 | Concern with cooperation | **.25** | .00 | **.35** | **-.24** | **.40** | | **.46** | **.19** | **.30** | **.45** | **.37** | -.05 | -.03 | **-.18** | .65 | 3.57 | .79 |
| 7 | Environmental concern | **.45** | .04 | **.49** | **-.26** | **.45** | **.39** | | **.30** | **.48** | **.53** | **.51** | **-.19** | .00 | **-.23** | .75 | 4.08 | .66 |
| 8 | PBC | .13 | .15 | **.39** | -.16 | **.18** | **.24** | **.27** | | **.33** | **.29** | **.24** | -.13 | -.05 | **-.21** | .86 | 4.20 | .79 |
| 9 | Perceived risk | **.35** | .02 | **.32** | **-.19** | **.27** | .17 | **.31** | **.22** | | **.49** | **.38** | **-.17** | .01 | -.08 | .93 | 4.73 | 1.20 |
| 10 | Anxiety | **.28** | .15 | **.67** | **-.41** | **.24** | **.28** | **.45** | **.25** | **.47** | | **.75** | **-.15** | -.01 | **-.20** | .88 | 3.76 | 1.03 |
| 11 | Distress | **.35** | .08 | **.59** | **-.36** | **.35** | **.32** | **.49** | **.24** | **.39** | **.69** | | **-.20** | -.14 | **-.34** | .80 | 3.68 | 1.23 |
| 12 | Political orientation | **-.30** | .04 | **-.33** | **.32** | -.05 | .00 | **-.28** | -.11 | -.13 | **-.21** | **-.28** | | **.14** | **.23** | - | 4.46 | 2.09 |
| 13 | Age | **-.15** | .07 | .01 | -.05 | -.11 | .06 | -.05 | -.01 | -.09 | -.08 | **-.16** | .10 | | **.16** | - | 28.03 | 11.06 |
| 14 | Gender | **-.14** | **-.18** | **-.15** | **.13** | -.10 | .00 | **-.26** | -.12 | -.12 | **-.20** | **-.39** | **.19** | .10 | | - | | |
| α | | .84 | .82 | .92 | .60 | .52 | .68 | .80 | .89 | - | .90 | .86 | - | - | - | | | |
| M | | 3.87 | 3.29 | 3.70 | 2.45 | 3.92 | 3.73 | 4.33 | 4.24 | 5.13 | 4.11 | 4.25 | 3.84 | 27.85 | - | | | |
| SD | | .61 | .62 | .89 | .66 | .61 | .77 | .66 | .78 | 1.20 | 1.06 | 1.26 | 1.61 | 9.67 | | | | |
| Theoretical range | | 1–5 | 1–4 | 1–5 | 1–5 | 1–5 | 1–5 | 1–5 | 1–5 | 1–6 | 1–6 | 1–6 | 1–11 | - | 1; 2; 3 | | | |

*Note.* PBC = perceived behavioral control. Bolded coefficients indicate statistical significance at p < .05. Descriptive statistics and correlations for Sample A are presented above, descriptive statistics and correlations for sample B below the diagonal.

= 0.71; $M_B$ = 3.87, $SD_B$ = 0.61). In both samples, general support for climate policy was positively associated with personal goal importance, perceived insufficiency in others' environmental behavior, concern with cooperation, environmental concern, and various risk perception indicators. Conversely, those who reported more frequent self-control struggle in the environmental domain and who self-identified as politically right leaning indicated lower general support for climate policy. Trait self-control and, in sample A, perceived behavioral control, were not significantly associated with policy support. Table 1 summarizes descriptive statistics and zero-order correlations for the main study variables in both samples.

**2.2.3 Predicting general support for climate policy.** We tested whether self-control variables and beliefs surrounding others' cooperation should be included in the best model predicting general support for behavioral climate policy, controlling for environmental attitudes, climate change-related risk perceptions, and sociodemographic variables. We applied 5-fold cross-validated elastic net regression to our list of variables to tune the model's hyperparameters. The results of the elastic net and the hyperparameter values resulting from 5-fold cross-validation are displayed in Table 2.

According to the elastic net algorithm, trait self-control, personal goal importance, self-control struggle, perceived insufficiency of others' environmental behavior, concern with cooperation, environmental concern, perceived behavioral control, perceived risk, anxiety, distress, political orientation, and gender predict general policy support.

To gauge effect sizes, we then ran linear regression models using the predictors identified by elastic net. Results show that those who report lower trait self-control and higher importance of climate mitigation, perceive others' environmental behavior as more insufficient, and are concerned with others' cooperation were more supportive of climate policy. Additionally, environmental concern and perceived risk of negative climate change consequences were positively associated with policy support, while those who report stronger perceived behavioral

**Table 2. Results of five-fold cross-validated elastic net regression and a multiple linear regression model predicting general climate policy support.**

| | General climate policy support | | | | | | | |
|---|---|---|---|---|---|---|---|---|
| | $\beta_{en}$ | $\beta$ | 95% CI | | SE | t | p | $\Delta R^2$ |
| Intercept | 0.05 | 0.07 | -0.06 | 0.20 | 0.06 | 1.10 | .271 | - |
| **Trait self-control** | **-0.06** | **-0.07** | **-0.14** | **0.00** | **0.04** | **-2.00** | **.046** | **.00** |
| **Importance** | **0.15** | **0.19** | **0.09** | **0.30** | **0.05** | **3.65** | **.000** | **.02** |
| Self-control struggle | -0.05 | -0.06 | -0.14 | 0.02 | 0.04 | -1.45 | .149 | .00 |
| **Perceived insufficiency** | **0.24** | **0.25** | **0.18** | **0.33** | **0.04** | **6.47** | **.000** | **.05** |
| **Concern with cooperation** | **0.08** | **0.08** | **0.00** | **0.16** | **0.04** | **2.05** | **.040** | **.00** |
| **Environmental concern** | **0.10** | **0.11** | **0.02** | **0.19** | **0.04** | **2.41** | **.016** | **.01** |
| **PBC** | **-0.06** | **-0.09** | **-0.16** | **-0.02** | **0.04** | **-2.38** | **.018** | **.01** |
| **Perceived risk** | **0.16** | **0.19** | **0.11** | **0.27** | **0.04** | **4.70** | **.000** | **.03** |
| Anxiety | -0.03 | -0.10 | -0.21 | 0.02 | 0.06 | -1.65 | .100 | .00 |
| Distress | 0.02 | 0.04 | -0.06 | 0.15 | 0.05 | 0.77 | .441 | .00 |
| **Political orientation** | **-0.09** | **-0.10** | **-0.17** | **-0.03** | **0.04** | **-2.81** | **.005** | **.01** |
| Age | . | . | . | . | . | . | . | . |
| Gender | -0.05 | -0.07 | -0.24 | 0.09 | 0.08 | -0.88 | .378 | .00 |
| Exp. group 1 | -0.02 | -0.05 | -0.21 | 0.12 | 0.08 | -0.57 | .570 | .00 |
| Exp. group 2 | . | 0.00 | -0.17 | 0.16 | 0.08 | -0.02 | .981 | .00 |
| Sample | . | 0.01 | -0.12 | 0.15 | 0.07 | 0.18 | .854 | .00 |
| Total adjusted $R^2$ | 0.32 | | | | | | | |
| $\lambda$ | 0.10 | | | | | | | |
| $\alpha$ | 0.086 | | | | | | | |

*Note.* $\beta_{en}$ = standardized regression coefficients of 5-fold cross-validated multiple regression model using elastic net; $\beta$ = standardized beta coefficients; $\Delta R^2$ = increase in $R^2$ resulting from the addition of the specified predictor; PBC = perceived behavioral control; exp. group = experimental group. Coefficients denoted as «.» were regularized out by elastic net. Bolded rows indicate statistically significant beta coefficients at $p < .05$.

control and those who report right-leaning political views are less supportive of climate policy. In total, the model accounted for 32% of the variance in general policy support. In terms of effect sizes, significant predictors ranged from $\Delta R^2$ = .00 (trait self-control, rounded down) to $\Delta R^2$ = .05 (perceived insufficiency; see Table 2). The results of both elastic net and OLS regression remain stable regarding the effects of our five main predictors of interest without the inclusion of covariates (i.e., environmental attitudes and beliefs, risk perception, and sociodemographic variables). When we do not apply our exclusion criteria (for ethical reasons, we still exclude minors), trait self-control is no longer a negative predictor of policy support ($p$ = .051, see Supplemental Analyses on the OSF, https://tinyurl.com/3jsk87re).

## 2.3 Discussion study 1

In this study, we explored whether individuals' own self-control success and the perceived insufficiency of others' environmental behavior would predict behavioral climate policy support beyond a range of established predictors. Trait self-control, personal goal importance, perceived insufficiency of others' environmental behavior, and concern with cooperation emerged as independent predictors of policy support, while self-control struggle in environmental behavior did not. Given that the effect of trait self-control is not entirely stable and relatively small, we are reluctant to offer an interpretation at this point. Notably, the observed effects emerged beyond those of environmental attitudes, risk perceptions, and sociodemographic variables, indicating incremental predictive value beyond these established predictors.

Given that previous literature has identified a link between trait self-control and environmental behavior [63–65], we retained trait self-control as a predictor for study 2 despite unstable effects in study 1. Study 2 further used an expanded measure of perceived insufficiency of others' environmental behavior with better internal consistency.

## 3. Study 2

### 3.1. Brief introduction

The goal of study 2 was to build on exploratory study 1 through replication and confirmation of our main effects of interest. To this end, we tested the hypotheses generated from study 1 and additionally included trait self-control based on theoretical considerations. We pre-registered these hypotheses (viewable on the OSF, https://tinyurl.com/3jsk87re) and separated them into primary hypotheses and secondary hypotheses, depending on the strength of evidence by study 1 data. Our primary hypotheses of interest stipulated that support for behavioral climate policy would be comparatively higher among those (1) reporting higher personal importance of climate change mitigation and (2) perceiving others' environmental behavior to be more insufficient. In a set of secondary hypotheses, we hypothesized that support for behavioral climate policy would be comparatively higher among those reporting (1) lower trait self-control and (2) higher concern with cooperation. In additional exploratory analyses, we also tested the applicability of these models to a more specific measure of policy support based on [50], which presents participants with example policies (pre-registered as a dependent variable for exploration). This allowed us to test whether our conceptual framework would extend to specific policy proposals as they might be voted on in real life. Previous work has linked descriptive and injunctive social norms to policy support [28, 30, 50, 66]. To test our predictors' incremental predictive power beyond social norms, we included them as additional control variables in this study. We additionally pre-registered the method of analysis, which we departed from to better meet the goal of this study. We describe this in more detail in the results section.

### 3.2. Method

**3.2.1. Sample.** For study 2, $N = 274$ adults residing in Germany completed the questionnaire. Out of this initial sample, we excluded $n = 4$ participants who failed the same one-item quality check employed in study 1. The final sample ($N = 270$) consisted of 53% women ($M$age = 45.63 years, $SD$age = 14.65 years, range = 18–78 years). Recruitment was conducted with the assistance of the market research organization *Respondi* (www.repondi.com/en; 2020/09/14–2020/09/22) and aimed to obtain a sample representative of the German general population in terms of gender and age (including gender distribution by age bracket). Concerning employment, 64% were employed full- or part-time. The remainder of participants indicated being retired (18%), in university (7%), unemployed (4%), homemakers (3%), on parental leave (3%), or none of these (1%). Our pre-registered target sample size of $N = 250$ (+ max. 10% for technical reasons in data collection via *Respondi*; see https://tinyurl.com/3jsk87re) was chosen to achieve 90% power for small to medium effect sizes for the maximum number of predictors included in exploratory analyses ($k = 15$) and informed by an a priori power calculation using G*Power 3.1. Participants were compensated €3.95 (USD 5) for their participation.

**3.2.2. Measures and procedure.** Participants completed a 15-minute online questionnaire on the German survey platform SoSci Survey (https://www.soscisurvey.de/en/index). Of the measures reported for study 1, study 2 included the following: general support for behavioral climate policy, perceived insufficiency of others' environmental behavior, concern with cooperation, personal goal importance, trait self-control, political orientation, gender, and age. As

declared in our preregistration, we included specific policy support as a secondary dependent measure, based on a recently published article [50]. We describe this measure below. The operationalization of study 1 variables remained the same, except perceived insufficiency of others' environmental behavior, for which we included three additional items to improve internal consistency. Given that these three additional items substantially improved internal consistency, we used the six-item scale instead of the previously used three-item scale, as pre-registered. According to the self-checklist provided by the university ethics board, an application for approval was not necessary.

*3.2.2.1. Support for specific behavioral climate policies.* Participants indicated the extent of their opposition versus support for ten proposed policies (e.g., "Regulate carbon dioxide (the primary greenhouse gas) as a pollutant"; "Provide tax rebates for people who purchase energy-efficient vehicles or solar panels"), which had been translated into German from [50]. The scale ranged from 1 (*do not agree at all*) to 5 (*completely agree*), with five reverse-scored items. We adapted two items, which referred to U.S.-specific policies, to make them applicable to the European context (see Supplementary Materials on the OSF, https://tinyurl.com/3jsk87re). Given the sufficient internal consistency (see "Descriptive Statistics and Zero-Order Correlations"), items were averaged with higher scores indicating higher support for these policies.

*3.2.2.2. Social norms.* Based on previous distinctions between descriptive and injunctive social norms [67], we asked participants to report both. To measure descriptive norms, we asked participants to estimate the percentage (0–100) of the German population making an effort to do something against climate change (*descriptive norm (general)*) and the percentage of family and friends making an effort to do something against climate change (*descriptive norm (close others)*). Please note that descriptive norms measure how *widespread* individuals perceive pro-environmental behaviors to be, without any evaluative component. Conversely, perceived insufficiency captures a *judgement of inadequacy* for climate change mitigation. As a proxy for injunctive norms, participants indicated to what extent they believed that their family and friends expect them to do something against climate change (1 = *not at all*– 5 = *very much*; -1 = *I don't know*).

## 3.3. Results

**3.3.1. Analyses.** To test the effects of perceived insufficiency of others' environmental behavior, concern with cooperation, importance, and trait self-control on general climate policy support, we ran a multiple regression model that also controlled for political orientation, gender, and age, as we did in study 1 (method of analysis presents a departure from the preregistration to better match the goal of the study). Given that we pre-registered hypothesis tests, as opposed to exploration, we employed confirmatory models (i.e., multiple regression) instead of exploratory variable selection (i.e., elastic net regression), as initially pre-registered. In our view, this provides the best fit with our goal of hypothesis testing. We do not report any additional exploratory analyses proposed in the preregistration as these are beyond the scope of this article. For our analyses, we employed R version 4.0.1 [68] and the *dplyr* [69], *psych* [70], and *magrittr* [71] packages for data cleaning.

**3.3.2. Descriptive statistics and zero-order correlations.** Descriptive statistics and bivariate correlations largely mirror those reported in study 1. Again, average general support for climate policy was moderate to high ($M = 3.35$, $SD = 0.88$) and support for specific climate policies was moderate ($M = 2.99$, $SD = 0.49$). General support for climate policy was moderately to highly positively correlated with personal goal importance, concern with cooperation, and support for specific climate policies. Again, those who perceived others' environmental behavior as more sufficient and self-identified as politically right leaning, as well as older adults

**Table 3. Summary of intercorrelations, internal consistencies, means, and standard deviations for main study variables.**

| | | 1 | 2 | 3 | 4 | 5 | 6 | 7 | 8 | 9 | 10 | 11 | 12 |
|---|---|---|---|---|---|---|---|---|---|---|---|---|---|
| 1 | General policy support | | | | | | | | | | | | |
| 2 | Specific policy support | **.62** | | | | | | | | | | | |
| 3 | Trait self-control | **-.09** | -.05 | | | | | | | | | | |
| 4 | Goal importance | **.62** | **.53** | .02 | | | | | | | | | |
| 5 | Perceived insufficiency | **.62** | **.55** | -.04 | **.61** | | | | | | | | |
| 6 | Concern with cooperation | **.45** | **.46** | -.03 | **.56** | **.65** | | | | | | | |
| 7 | Descriptive norms (general) | **-.22** | **-.18** | .07 | .01 | **-.36** | **-.20** | | | | | | |
| 8 | Descriptive norms (close others) | -.08 | -.01 | **.18** | **.17** | **-.14** | -.05 | **.73** | | | | | |
| 9 | Injunctive norms | **.22** | **.25** | **.15** | **.58** | **.26** | **.34** | **.27** | **.44** | | | | |
| 10 | Political orientation (1 = *left*– 10 = *right*) | **-.32** | **-.28** | .06 | **-.22** | **-.26** | -.12 | **.17** | .06 | -.03 | | | |
| 11 | Age | **-.18** | **-.17** | **.26** | -.05 | -.11 | .05 | .05 | .06 | **.16** | .11 | | |
| 12 | Gender (1 = *female*; 2 = *male*) | .04 | .09 | -.05 | .05 | .05 | .02 | .09 | .08 | .13 | **.15** | -.02 | |
| α | | .93 | .78 | .82 | .93 | .86 | .80 | - | - | - | - | - | - |
| M | | 3.35 | 2.99 | 3.37 | 3.56 | 3.65 | 3.84 | 45.10 | 50.92 | 3.42 | 5.80 | 45.63 | - |
| SD | | .88 | .49 | .59 | .92 | .74 | .83 | 20.19 | 25.96 | 1.97 | 1.95 | 14.65 | - |
| Theoretical range | | 1–5 | 1–4 | 1–5 | 1–5 | 1–5 | 1–5 | 1–100 | 1–100 | 1–5 | 1–10 | - | 1; 2 |

*Note*. Bolded coefficients indicate statistical significance at p < .05.

indicated lower general support for climate policy. Zero-order correlations with support for specific climate policies are comparable, with slightly smaller effect sizes. Correlations between descriptive and injunctive norms and perceived insufficiency of others' behavior were small to moderate, $r$s = |.14|—|.36| (absolute values). Additionally, correlations with policy support were considerably weaker (or non-significant) for norm variables compared to insufficiency of others' behavior. Table 3 summarizes descriptive statistics and zero-order correlations for the main study variables.

**3.3.3. Predicting general climate policy support.** Multiple linear regression models show that general policy support is predicted by personal goal importance and perceived insufficiency of others' environmental behavior, replicating study 1 results, and providing evidence for our primary hypotheses. In other words, those who reported climate change mitigation as a highly important personal goal and who perceived others as not contributing sufficiently to climate change mitigation were more in favor of government-imposed climate policy in general. Notably, these variables' effects emerged beyond those of descriptive and injunctive social norms, which have previously been linked to policy support, and collinearity checks returned acceptable variance inflation factors (VIF) well below the standard cut-off of VIF = 5 [72], e.g. $VIF_{norms\ population}$ = 2.73 and $VIF_{norms\ (family\ and\ friends)}$ = 2.69. Our secondary hypotheses were not supported by the data, such that neither trait self-control nor concern with cooperation significantly predicted policy support. In total, the model accounted for 56% of the variance in general policy support. In terms of effect sizes, significant predictors ranged from $\Delta R^2$ = .02 (perceived insufficiency) to $\Delta R^2$ = .09 (importance; see Table 4). These effect sizes indicate the variable's incremental predictive contribution to the model and therefore do not include shared variance between predictors. The results remained stable with the inclusion of all cases (though we still exclude minors for ethical reasons) and the exclusion of covariates.

**3.3.4 Predicting support for specific climate policies.** Again, personal importance of climate change mitigation and perceived insufficiency of others' environmental behavior emerge as the two strongest predictors of support for specific policies, mirroring models predicting

**Table 4. Results of a regression model predicting general climate policy support.**

| | General climate policy support | | | | | | |
| --- | --- | --- | --- | --- | --- | --- | --- |
| | β | 95% CI | | SE | t | p | $\Delta R^2$ |
| Intercept | -0.08 | -0.22 | 0.06 | 0.07 | -1.07 | .287 | - |
| Trait self-control | -0.04 | -0.14 | 0.07 | 0.05 | -0.68 | .499 | .00 |
| **Importance** | **0.49** | **0.34** | **0.65** | **0.08** | **6.27** | **.000** | **.09** |
| **Perceived insufficiency** | **0.26** | **0.10** | **0.42** | **0.08** | **3.24** | **.001** | **.02** |
| Concern with cooperation | 0.04 | -0.09 | 0.17 | 0.07 | 0.57 | .567 | .00 |
| Descriptive norms (general) | -0.10 | -0.25 | 0.05 | 0.08 | -1.26 | .211 | .00 |
| Descriptive norms (close others) | -0.02 | -0.17 | 0.13 | 0.08 | -0.26 | .794 | .00 |
| Injunctive norms | -0.12 | -0.25 | 0.02 | 0.07 | -1.72 | .088 | .03 |
| **Political orientation** | **-0.11** | **-0.21** | **-0.01** | **0.05** | **-2.21** | **.028** | **.04** |
| Gender | 0.11 | -0.09 | 0.31 | 0.10 | 1.08 | .280 | .00 |
| Age | -0.11 | -0.21 | 0.00 | 0.05 | -1.96 | .052 | .01 |
| Total adjusted $R^2$ | 0.56 | | | | | | |

*Note*. B = standardized beta coefficients; $\Delta R^2$ = increase in $R^2$ resulting from the addition of the specified predictor. Bolded rows indicate statistically significant beta coefficients at p < .05.

general policy support. Additionally, support for specific policies increases alongside concern with others' cooperation. Again, those reporting right-leaning political orientation tend to be less supportive. Neither descriptive nor injunctive social norms emerged as significant predictors of policy support. Coefficient estimates from multiple linear regression models are displayed in Fig 1 for both studies and outcome variables to facilitate comparisons between effect sizes across samples and outcome variables.

In total, the model accounts for 47% of the variance in support for specific climate change mitigation policies. In terms of effect sizes, significant predictors ranged from $\Delta R^2$ = .00 (concern with cooperation, rounded down) to $\Delta R^2$ = .09 (importance; see Table 5). Again, these effect sizes indicate the variable's incremental predictive contribution to the model. The results of both models remain unchanged when we do not apply our exclusion criteria (for ethical reasons, we still exclude minors). Without the inclusion of covariates, the results remain the same except for concern for cooperation, which is no longer a significant predictor (see Supplemental Analyses on the OSF, https://tinyurl.com/3jsk87re).

### 3.4. Discussion study 2

Having established the incremental predictive power of a set of self-control and cooperation variables beyond established predictors in study 1, we systematically tested their association with policy support in confirmatory analyses in study 2. We further expanded on study 1 by including descriptive and injunctive social norms, which have previously been linked with policy support. Results indicate that personal goal importance, perceived insufficiency in others' environmental behavior, and, to a lesser extent, concern with cooperation predict policy support and, indeed, provide incremental predictive value beyond social norms.

### 4. General discussion

Previous work has identified a range of psychological factors that predict support for behavioral climate change policy [8, 50, 66, 73]. However, this research is the first to put forward a conceptualization of climate change mitigation as a cooperative self-control problem. The current study provides evidence that perceiving others as insufficiently cooperative is associated

**Coefficient Sizes Across Studies**

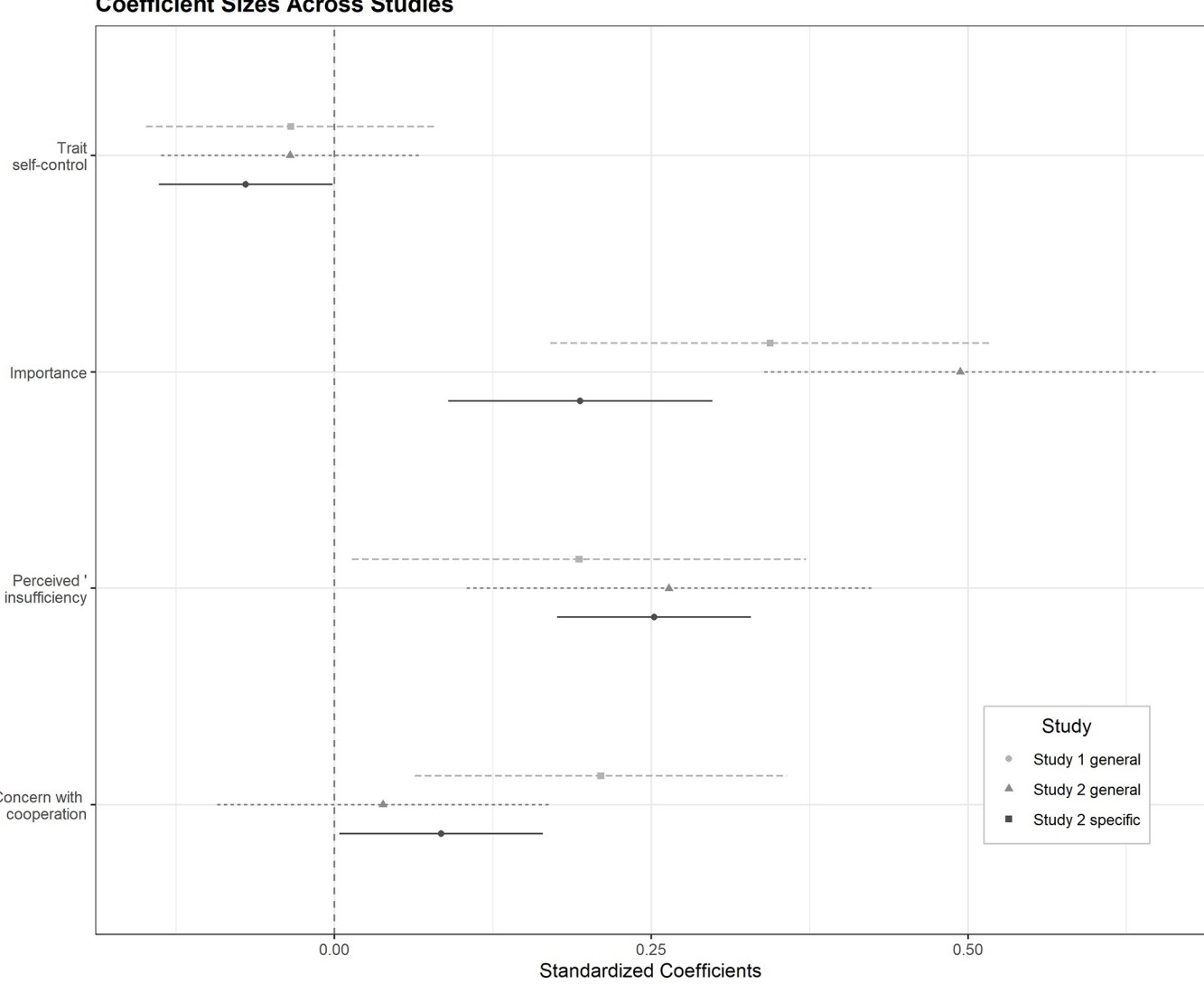

**Fig 1. Coefficient sizes and associated 95% confidence intervals across both studies and outcome variables for predictors of interest.** *Note.* Polit. orientation = political orientation, general = general support for climate policy, specific = support for specific climate policies. In study 1, age was not selected by elastic net and is, therefore, not included here.

with greater policy support, as is personal goal importance, a key component of the self-control process. Notably, we found mixed evidence that policy support emerges as a function of individuals' own general difficulty in self-regulation *(trait self-control)* and no evidence specifically in the environmental domain *(self-control struggle)*. This discrepancy between how policy may be instrumentalized to restrict own versus others' behavior is in line with the third person effect, which describes the phenomenon that individuals see others as more malleable by social influence [39] and paternalistic aid [74] than themselves.

In study 1, we show that trait self-control, goal importance, perceived insufficiency of others' environmental behavior, and concern with cooperation emerge as important independent predictors of policy support, providing incremental predictive value beyond an established set of psychological and sociodemographic predictors. In study 2, we systematically test pre-registered hypotheses regarding the associations of self-control and beliefs surrounding others'

**Table 5. Results of a multiple linear regression model predicting support for specific climate policies.**

| | Specific climate policy support | | | | | | |
|---|---|---|---|---|---|---|---|
| | β | 95% CI | | SE | t | p | ΔR² |
| Intercept | -0.08 | -0.24 | 0.07 | 0.08 | -1.05 | .296 | - |
| Trait self-control | -0.03 | 0.01 | 0.37 | 0.06 | -0.59 | .555 | .00 |
| **Importance** | **0.34** | **0.06** | **0.36** | **0.09** | **3.89** | **.000** | **.09** |
| **Perceived insufficiency** | **0.19** | **-0.15** | **0.08** | **0.09** | **2.11** | **.036** | **.02** |
| **Concern with cooperation** | **0.21** | **0.17** | **0.52** | **0.07** | **2.80** | **.006** | **.00** |
| Descriptive norms (general) | -0.12 | -0.29 | 0.05 | 0.09 | -1.35 | .179 | .00 |
| Descriptive norms (close others) | 0.07 | -0.10 | 0.24 | 0.09 | 0.85 | .398 | .00 |
| Injunctive norms | -0.05 | -0.20 | 0.10 | 0.08 | -0.65 | .518 | .03 |
| **Political orientation** | **-0.12** | **-0.23** | **-0.01** | **0.06** | **-2.19** | **.029** | **.04** |
| Gender | 0.08 | -0.14 | 0.31 | 0.11 | 0.74 | .459 | .00 |
| Age | -0.11 | -0.22 | 0.01 | 0.06 | -1.74 | .083 | .01 |
| Total adjusted R² | 0.47 | | | | | | |

*Note*. β = standardized beta. ΔR² = increase in R² resulting from the addition of the specified predictor. Bolded rows indicate statistically significant beta coefficients at p < .05.

cooperation with policy support based on study 1 findings. Our primary hypotheses, which specify that those who hold climate change mitigation as a highly important goal and perceive others as insufficiently cooperative in climate change mitigation, were fully supported by the data. These associations generalized to a measure of support for specific policies. Our secondary hypotheses, which specified that those lower in trait self-control and more concerned with others' cooperation in climate change mitigation would be more supportive of policy, were not supported. However, analyses indicate that those who are more concerned with equitable contributions in climate change mitigation (*concern with cooperation*) are more supportive of *specific* policy proposals (rather than general policy support).

We invite subsequent work to build on these results, particularly regarding the regulation of uncooperative others through behavioral policy. It would be insightful to investigate generalization across other collective action problems and across cultures, and to establish the boundary conditions under which the effect emerges. For instance, future work might test whether the present findings generalize to other collective action domains (e.g., biodiversity loss or antimicrobial resistance, see [5]) or to specific problems in these domains (e.g., choice of transportation leading to congestion). From the present studies, it remains unclear whether our findings reflect a fundamental divergence between individuals' preferences for regulating their own versus others' behavior in collective action problems [75]. Follow-up work might, therefore, study how awareness of collective interdependence in goal attainment moderates these results. Given the tendency for individuals to engage in costly punishment when they perceive insufficient cooperation [34, 76], potential mediators of the present results include the desire to punish uncooperative others versus alignment with the collective goal (in this case, climate change mitigation). Building on Ostrom's work on social dilemmas [77], the observed results could also be interpreted as contingent cooperation, such that individuals will cooperate on the condition that others do. Policy governs all citizens and may therefore constitute a means of enforcing cooperation. Future experimental work might compare the effects of the proposed candidate mechanisms (i.e., goal advancement, social punishment, contingent cooperation).

Regarding individuals' own shortcomings in self-controlled behavior, it is conceivable that support for external regulation of individual behavior may emerge as a function of self-control failure under specific conditions. For instance, individuals may prefer regulation of their own behavior to be more behaviorally proximal, such as local or institutional (vs. national government) regulation or social pre-commitment devices. Future studies may also consider the extent to which individuals are aware that punishment of the collective entails regulation of their own behavior, given that it is applied to all members of society, or whether the effect differs in magnitude according to individuals' own self-control success or goal importance. Given that vastly larger sample sizes are required to detect interaction effects [78], we call for future sufficiently powered studies to test potential moderators.

Given the limited availability of resources, such as time and money, policymakers must focus their efforts on the most critical factors [50] when campaigning for political change and communicating policy options to the public. The present studies expand on past work by integrating novel, theoretically derived with previously identified psychological predictors of policy support into one statistical model, gauging each variable's independent contribution. These findings help synthesize extant work, which has primarily investigated psychological predictors of climate policy in isolation. Separating exploration and confirmation into two studies helps to prevent myopic focalization with a new conceptual approach by taking into account previous findings and showing our target predictors' relevance beyond these variables.

### 4.1. Theoretical and practical implications

Across samples, personal goal importance emerged as one of the variables with the greatest predictive strength. These findings align with fundamental motivation principles that point to goal commitment, which is conceptually similar to our operationalization of personal goal importance, as the required first step in initiating goal striving. According to the Rubicon model of action phases [79], commitment to a personal goal propels action oriented toward attaining this goal. Integrative self-control theory [10] similarly defines personal goal importance as a critical driver of the self-control process, which aligns behavior with the target goal state. Recent work has demonstrated that it is possible to capitalize upon the central role of goal importance in environmental behavior, showing that activating personal goals by translating choice attributes into goal-relevant information increases pro-environmental decision-making [80]. Policymakers might, therefore, consider highlighting the importance of proposed policies for climate protection to link to pre-existing personal climate change mitigation goals.

Perceived insufficiency of others' environmental behavior also consistently emerged as a strong predictor of climate policy support. In line with social dilemma research, which indicates a tendency to coerce cooperation and punish defection to increase cooperation [35, 36, 81, 82], our findings, therefore, suggest a noteworthy social component in collective goal-striving. Importantly, our conceptualization of perceived insufficiency of others' environmental behavior moves beyond previous work on social norms by reflecting normative beliefs about the *adequacy* of others' environmentally relevant actions, instead of merely describing the status quo (i.e., descriptive norms). Past work has focused on the link between descriptive social norms (e.g., do other people support policy?) and own policy support, or descriptive social norms regarding environmental behavior (e.g., do others act pro-environmentally?) and own environmental behavior. Our work investigates the link between perceptions of the sufficiency of others' environmental behavior and individuals' support for behavioral policy to address climate change. In line with this distinction, the reported results remain stable with the inclusion of descriptive and injunctive social norms. Based on perceptions that others are not doing

enough to combat climate change, individuals may leverage behavioral policy as an enactment constraint (see [10]) to increase cooperation in others, and consequently, to aid climate change mitigation. However, this assumption remains to be tested by future work. We want to stress that these results should not be taken as grounds for communicating misinformation to the public as a means of increasing policy support. Indeed, false information will undermine trust in government and science and, ultimately, destabilize democracies and their policy instruments. Instead, we suggest providing accurate information on current shortcomings in environmental behavior, tailoring policy communication to social networks to engage well-connected "key" individuals [83]. Past work has demonstrated that simple changes in framing, especially leveraging social influence, impacts real-world voting behavior [84], underlining the importance of communication approaches.

## 4.2. Strengths

To date, the field of environmental psychology has focused on hypothesis-testing research, with few studies employing a data-driven approach. While hypothesis-testing approaches are necessary steps in the scientific process, we join IJzerman and colleagues [53] in highlighting exploratory work as an essential building block of discovery. In fact, the problem of overfitting (that is, specifying models that do not generalize beyond the data they closely represent) contributes to false-positive findings and thereby exacerbates the replication crisis in psychology. In this set of studies, we have capitalized upon the strengths of both exploratory and confirmatory analysis approaches. In study 1, we delineated *independent* effects of self-control and cooperation variables on policy support beyond the effects of previously identified predictors. We then showed that these associations hold in pre-registered confirmatory tests of the model in study 2. Further, the strongest predictors of policy support emerged in samples from both a student and general population, providing evidence for generalizability beyond the highly educated, young, and politically liberal.

## 4.3. Limitations and future work

A central limitation of this work concerns the data's correlational nature, which prevents us from delineating causative associations with policy support. We further cannot speak to functional associations between predictors (e.g., potential paths from risk judgment to policy support via perceived insufficiency of others' environmental behavior) as our data are cross-sectional. It is for future experimental and intensive longitudinal work to deduce sequential associations between predictors, which will allow researchers to build process models of policy support. This model may be based on existing process models of self-control (e.g., [10]) but should additionally include variables that reflect the social aspects found in this research (e.g., insufficiency of others' behavior).

   Further, these data are based upon self-report. Future work might implement computerized game environments, as seen in classic social dilemma studies, that allow for the manipulation of behavioral difficulty and other players' cooperation. Studies may also wish to capitalize upon large-scale panel data that include referendum voting choices, though such panels rarely include psychological variables, making it challenging to estimate integrative models as presented in this article. Lastly, our work cannot be considered a comprehensive account of psychological policy support predictors due to feasibility constraints. However, our initial set of predictor variables is grounded in a theoretically and empirically informed selection, and we find effects of cooperation and self-control components beyond those of established predictors (e.g., environmental attitudes, risk perception, political orientation), indicating their incremental predictive value.

Future work might test whether communication on current insufficiencies in environmental behavior can be combined with collective efficacy or shared identity messages, which have been shown to promote pro-environmental behavior [85]. Indeed, previous authors (e.g., [86]) have argued that "social nudges," that is, nudges that inform and raise normative concerns about others' behavior, are among the most effective in inciting desirable behavior change. This reiterates the importance of engaging the social context in climate policy communication.

## 5. Conclusion

Using a combination of exploratory and confirmatory analyses, we found evidence that those who hold climate change mitigation as an important personal goal and who perceive others as insufficiently cooperative in climate change mitigation are relatively more supportive of climate policy. We find this association across three samples, first in a two-sample exploratory study using mixed student and community samples, then in a pre-registered replication study using an age- and gender-representative sample from the general population. These findings align with our framing of climate mitigation as a cooperative self-control problem based on extant work [3, 11, 15]. Notably, the reported effects emerged beyond those of other well-established predictors, such as social norms, environmental attitudes, affective and analytic risk perception, political orientation, gender, and age. Future work can test whether the wish to control one's own or others' collectively relevant behavior indeed a key driver of leveraging behavioral policy for pro-environmental behavior enforcement. Additional experimental and intensive longitudinal work may serve to identify functional and sequential associations between the identified predictors, and to map how people's social and physical environments shape preference for external regulation.

## Author Contributions

**Conceptualization:** Charlotte A. Kukowski, Katharina Bernecker, Veronika Brandstätter.

**Data curation:** Charlotte A. Kukowski, Leoni von der Heyde, Margarete Boos.

**Formal analysis:** Charlotte A. Kukowski.

**Investigation:** Charlotte A. Kukowski.

**Methodology:** Charlotte A. Kukowski, Katharina Bernecker.

**Project administration:** Charlotte A. Kukowski.

**Resources:** Margarete Boos, Veronika Brandstätter.

**Supervision:** Katharina Bernecker, Margarete Boos, Veronika Brandstätter.

**Visualization:** Charlotte A. Kukowski.

**Writing – original draft:** Charlotte A. Kukowski.

**Writing – review & editing:** Katharina Bernecker, Leoni von der Heyde, Margarete Boos, Veronika Brandstätter.

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
