## [Decision Letter · Decision Letter 0]

28 Feb 2022

PONE-D-21-40332Climate policy support as a tool to control others’ (but not own) environmental behavior?PLOS ONE

Dear Dr. Kukowski,

Thank you for submitting your manuscript to PLOS ONE. After careful consideration, we feel that it has merit but does not fully meet PLOS ONE’s publication criteria as it currently stands. Therefore, we invite you to submit a revised version of the manuscript that addresses the points raised during the review process.

I have received two reviews and carefully read your manuscript myself. I apologize for not sending my editorial decision before, but a third reviewer had accepted to evaluate your manuscript and I was waiting for their review to arrive. To allow your manuscript to procede the next stages toward publication without further delay, I have now chosen to make my decision based on the two existing reviews. I take the opportunity to thank the reviewers for their very helpful and constructive contribution.

Both I and the reviewers have highly appreciated your work: The research addresses a timely and highly relevant issue, the manuscript is well written, the research incorporates the best practices for transparency, robustness and replicability.

Before the manuscript can be accepted for publication, however, I ask you to address the points raised by the reviewers. In your response letter, for each point raised by the reviewers, please indicate clearly how you have addressed it in the manuscript and where in the manuscript the changes have been incorporated / the reason why you have preferred not to address the point. 

We look forward to receiving your revised manuscript.

Kind regards,

Cristina Zogmaister, Ph.D.

Academic Editor

PLOS ONE

Journal Requirements:

Reviewers' comments:

Reviewer's Responses to Questions

**Comments to the Author**

1. Is the manuscript technically sound, and do the data support the conclusions?

Reviewer #1: Yes

Reviewer #2: Yes

2. Has the statistical analysis been performed appropriately and rigorously? 

Reviewer #1: I Don't Know

Reviewer #2: Yes

3. Have the authors made all data underlying the findings in their manuscript fully available?

Reviewer #1: Yes

Reviewer #2: Yes

4. Is the manuscript presented in an intelligible fashion and written in standard English?

Reviewer #1: Yes

Reviewer #2: Yes

5. Review Comments to the Author

Reviewer #1: This paper generates and then tests hypotheses regarding the expected importance of both beliefs about one’s own self-control related to environmental conservation behavior and also beliefs about the likely actions of others to improve their conservation behavior on support for new climate mitigation policies. The results are consistent with the idea that greater worries about the insufficiency of the actions of others to improve their environmental behaviors are associated with greater support for climate mitigation policies, but not for the hypothesis about concern about self-control predicting greater policy support. The authors portray this conclusion as more evidence for thinking about climate change as a “cooperative self-control” problem where policy support should be affected in part by beliefs about the likely actions of others to cooperate in addressing the problem.

The paper’s findings are novel and important both theoretically and in terms of policy making, which is an important strength. The research design is also well structured and very thorough, with two separate surveys to distinguish hypothesis generating from hypothesis testing data, making the results more persuasive. The theoretical grounding of the key variables is also carefully explained, including for the key control variables.

One exception to this is a potential point of confusion between two variables—the hypothesized variable of “perceived insufficiency of others pro-env behavior” with the later descriptive norm variable used as a control which asks respondents to estimate the percentage of the population “making an effort to do something against climate change” (p. 22). This seems nearly indistinguishable, to me, from the “perceived insufficiency” variable that is the research focus (which relies on a similar norm-like perception of the behaviors of others). Given the use of the descriptive norm as a control in study 2, I think the authors need to discuss this potential overlap, both theoretically (how are these variables different?) and also in terms of the results (e.g., how high is the collinearity between the two variables?). The authors seem to want to dismiss norms as a cause here, when I would interpret their primary hypothesis as very close to a descriptive norm. This theoretical ambiguity should be clarified.

I also have some concerns with the interpretation of the results in the paper, especially with regard to recommendations for policy makers. Although the authors want to preserve the possibility that support for policies might still be caused by a lack of personal self-control, I find their results more consistent with a robust literature related to the notion of “contingent consent” in cooperative dilemmas. The importance of beliefs about how others will contribute to an environmental goal in support for government coercion on that issue is very consistent with a long line of research documenting this kind of thinking in other collective action dilemmas dating back to the work of Elinor Ostrom (1990), such as conscription (Levi 1997) or taxes (Scholtz and Lubell 1998) or even energy conservation (Bolsen 2011). In other words, there is a robust literature on people wanting to contribute to a public good, but only if they are convinced others are also contributing their fair share. This work is not considered in the article, yet it fits the results very well. I would encourage the authors to consider this notion of contingent consent as a key implication of their findings.

On a related note, the authors cite the strength of personal goal importance as a predictor of policy support in recommending policymakers make the environmental impacts of their policies as “visible as possible” (p. 31). I do not think these results justify this recommendation. Conceptually, if personal goal importance is the best predictor of policy support, that does not mean that worries about a policy’s connection to the env goal are the biggest threat to support. Indeed, with the findings about worries about the behavior of others, we can interpret these results as or more intuitively (in my view) as indicating that fear of a collective action failure is the biggest threat to policy support at the margin. In other words, even if I fully understand the environmental impacts of the policy, I may not support it if I don’t think it will change behavior among most or all other members of society. This might well be the more important policy implication of the paper, and the authors should consider these possibilities more carefully in writing up their “practical” implications for policymaking.

Reviewer #2: This is one of the best manuscripts that I've seen in several decades. The literature review is excellent. The integration of key social psychological theory and findings with the specific goals of this pair of studies is creative and right on target. It is exactly the kind of research that I've been urging social psychology students and colleagues to do for over a decade. Here are a few suggestions for minor changes:

1. Line 271. Include one sample item of the Personal Goal Importance scale in the text.

2. Did the authors consider the possibility of moderation effects? For example, is it possible that trait self-control interacts with perceived insufficiency of others' behavior? Exploratory analyses of this type might prove interesting.

3.p. 30. Another potentially useful follow-up study would be to replicate this study across a wide range of countries/cultures.

4. One of the major problems in the climate change/psychology domain is the relative isolation of scholars in rather narrow towers of specialization. I first became aware of this more that a decade ago when key editors of psychology works were largely uninterested in publishing works that integrated both the geological/physical aspects of climate change with the role played by psychological factors. I offer two specific citations for the authors' consideration, one of which could be seen as quite self serving. This first is a new monograph (released this week) that I co-authored:

Miles‐Novelo, A. and Anderson, C. A. (2022). Climate Change and Human Behavior: Impacts of a Rapidly Changing Climate on Human Aggression and Violence. Cambridge, UK: Cambridge University Press. https://www.cambridge.org/core/elements/abs/climate-change-and-human-behavior/F64471FA47B8A6F5524E7DDDDE571D57

This monograph is free to download from the above link, until Feb. 21. It also can be downloaded from my web site:

www.CraigAnderson.org

The second one is a recent article:

Barlett, C. P., DeWitt, C. C., Madison, C. S., Heath, J. B., Maronna, B., & Kirkpatrick, S. M. (2020). Hot temperatures and even hotter tempers: Sociological mediators in the relationship between global climate change and homicide. Psychology of Violence, 10(1), 1-7. http://dx.doi.org/10.1037/vio0000235

Please note that I am urging the authors to add either reference to this paper. The present version is excellent with or without describing either of these.

Craig

Craig A. Anderson, Ph.D.

Editor, Aggressive Behavior

Distinguished Professor of Psychology

Iowa State University

caa@iastate.edu

6. PLOS authors have the option to publish the peer review history of their article (what does this mean?). If published, this will include your full peer review and any attached files.

Reviewer #1: No

Reviewer #2: **Yes: **Craig A. Anderson

---

## [Author Response · Author response to Decision Letter 0]

8 Apr 2022

[Please see attached Word document for the correctly formatted rebuttal letter, including citations and relevant figures.]

Dear Dr. Zogmaister, dear Dr. Anderson, dear Reviewer,

We gratefully acknowledge the time and effort that you have invested into reading and commenting on our manuscript and would like to thank you for your highly constructive input. In our revisions and replies to your comments, we have aimed to address all points adequately and in sufficient detail. 

Please find below our detailed responses. Reviewer comments are fully quoted in italics, while author replies are written in plain text. Modifications to the manuscript are indented in plain text and preceded by an arrow (→).

Review Comments to the Author

Reviewer #1:

This paper generates and then tests hypotheses regarding the expected importance of both beliefs about one’s own self-control related to environmental conservation behavior and also beliefs about the likely actions of others to improve their conservation behavior on support for new climate mitigation policies. The results are consistent with the idea that greater worries about the insufficiency of the actions of others to improve their environmental behaviors are associated with greater support for climate mitigation policies, but not for the hypothesis about concern about self-control predicting greater policy support. The authors portray this conclusion as more evidence for thinking about climate change as a “cooperative self-control” problem where policy support should be affected in part by beliefs about the likely actions of others to cooperate in addressing the problem.

The paper’s findings are novel and important both theoretically and in terms of policy making, which is an important strength. The research design is also well structured and very thorough, with two separate surveys to distinguish hypothesis generating from hypothesis testing data, making the results more persuasive. The theoretical grounding of the key variables is also carefully explained, including for the key control variables.

Thank you very much for your positive feedback on our manuscript. We appreciate your constructive criticism and your encouragement regarding the value and execution of the manuscript and would like to thank you for your time and effort.

1. One exception to this is a potential point of confusion between two variables—the hypothesized variable of “perceived insufficiency of others pro-env behavior” with the later descriptive norm variable used as a control which asks respondents to estimate the percentage of the population “making an effort to do something against climate change” (p. 22). This seems nearly indistinguishable, to me, from the “perceived insufficiency” variable that is the research focus (which relies on a similar norm-like perception of the behaviors of others). Given the use of the descriptive norm as a control in study 2, I think the authors need to discuss this potential overlap, both theoretically (how are these variables different?) and also in terms of the results (e.g., how high is the collinearity between the two variables?). The authors seem to want to dismiss norms as a cause here, when I would interpret their primary hypothesis as very close to a descriptive norm. This theoretical ambiguity should be clarified.

Thank you for pointing out this potential point of conceptual ambiguity. We agree that it is important to clarify how these two variables differ conceptually and empirically. In short, social norms capture how widespread pro-environmental behavior is perceived to be, both in the German general population and in family and friends. Norms are defined as «belief systems about how (not) to behave that guide behaviour, but without the force of laws, and reflect group members’ shared expectations about typical or desirable activities» (Hewstone, et al., 2015, p. 604). Crucially, perceived insufficiency of others’ pro-environmental behavior captures a judgement of how adequate others’ behavior is for climate change mitigation (i.e., are others doing enough). While perceptions of how widespread pro-environmental behaviors are in the population and in family and friends should inform judgements of these behaviors’ adequacy for climate change mitigation, the valuative component is unique to the perceived insufficiency variable. This is indeed what we see both in the bivariate correlations between norms and insufficiency of others’ behavior, which are small to moderate, and in the regression results, which suggest an incremental predictive value of perceived insufficiency beyond social norms.

Hewstone, M., Stroebe, W., & Jonas, K. (2015) (Eds), An introduction to social psychology. Wiley.

To further address potential multicollinearity concerns, we ran model checks using the performance package in R. Variance inflation factors (VIF) are all well below 5 (cit.), with the relevant norm variables and perceived insufficiency ranging between 2 and 3. 

> performance::check_collinearity(mymodel)

# Check for Multicollinearity

Low Correlation

 Term VIF Increased SE Tolerance

 trait_sc 1.16 1.08 0.86

 goal_importance 2.83 1.68 0.35

 insufficiency 3.01 1.73 0.33

 concern_cooperation 2.07 1.44 0.48

 dnorm_population 2.73 1.65 0.37

 dnorm_fam_friends 2.69 1.64 0.37

 inj_norm 2.02 1.42 0.49

 political_orient 1.14 1.07 0.88

 gender 1.10 1.05 0.91

 Age 1.18 1.09 0.85

The results of collinearity checks using the performance::check_model() function in R are additionally visualized below:

Model results remain largely unchanged when excluding any one or all of the norm variables, with a moderate increase in the effect size of perceived insufficiency from β = 0.19 to β = 0.27 when excluding both descriptive norm measures and injunctive norms. This change likely reflects the small to moderate shared variance with norm indicators, which is to be expected. 

→ Please note that descriptive norms measure how widespread individuals perceive pro-environmental behaviors to be, without any evaluative component. Conversely, perceived insufficiency captures a judgement of inadequacy for climate change mitigation. (p. 22)

→ Correlations between descriptive and injunctive norms and perceived insufficiency of others’ behavior were small to moderate, rs = |.14| - |.36| (absolute values). Additionally, correlations with policy support were considerably weaker (or non-significant) for norm variables compared to insufficiency of others’ behavior. (p. 23)

→ Notably, these variables' effects emerged beyond those of descriptive and injunctive social norms, which have previously been linked to policy support, and collinearity checks returned acceptable variance inflation factors (VIF) well below the standard cut-off of VIF = 5 (70), e.g. VIFnorms population = 2.73 and VIFnorms (family and friends) = 2.69. (p. 25)

70. Akinwande MO, Dikko HG, Samson A. Variance Inflation Factor: As a Condition for the Inclusion of Suppressor Variable(s) in Regression Analysis. Open J Stat. 2015;05(07):754–67.

2. I also have some concerns with the interpretation of the results in the paper, especially with regard to recommendations for policy makers. Although the authors want to preserve the possibility that support for policies might still be caused by a lack of personal self-control, I find their results more consistent with a robust literature related to the notion of “contingent consent” in cooperative dilemmas. The importance of beliefs about how others will contribute to an environmental goal in support for government coercion on that issue is very consistent with a long line of research documenting this kind of thinking in other collective action dilemmas dating back to the work of Elinor Ostrom (1990), such as conscription (Levi 1997) or taxes (Scholtz and Lubell 1998) or even energy conservation (Bolsen 2011). In other words, there is a robust literature on people wanting to contribute to a public good, but only if they are convinced others are also contributing their fair share. This work is not considered in the article, yet it fits the results very well. I would encourage the authors to consider this notion of contingent consent as a key implication of their findings.

We absolutely agree that supporting policy – which will affect everyone, including the individual in question - when one perceives others not to be doing enough to combat climate change makes it more likely that others will contribute in the future. We agree that Elinor Ostrom’s work (and the body of literature descendant from it) is highly relevant to research on collective action dilemmas. We now cite her work in the introduction section when discussing collective action dilemmas:

→ However, it remains unclear how perceptions of the insufficiency and importance of others’ environmental behavior, that is, whether others are perceived to be doing enough to address climate change, associates with support for policy. Classic research on collective action problems suggests that individuals will contribute to a common resources if others do as well, a phenomenon known as contingent consent (32,33). Individuals are therefore motivated to align others’ behavior with the collective goal and to punish those who do not cooperate, even at a cost to themselves (34), across cultures (35). (p. 6)

32. Ostrom E. Collective action and the evolution of social norms. J Econ Perspect. 2000;14(3):137. 

33. Kopelman S, Weber JM, Messick DM. Factors influencing cooperation in commons dilemmas: A review of experimental psychological research. In: Ostrom E, Dietz T, Dolsak N, Stern PC, Stonich S, Weber EU, editors. The drama of the commons. National Academy Press; 2002. p. 113–56. 

Indeed, the variable referred to as “concern with cooperation” in our paper shares conceptual similarity to contingent consent / conditional cooperation in that is captures concern with fair and equal contributions in addressing climate change. However, this variable adds little explanatory value to the model, returning a small effect size in Study 1, β = .08, 95% CI [.00, .16], p = .040, and non-significant coefficient in Study 2, β = .04, 95% CI [-.09, .17], p = .567, and we did not test mechanisms linking perceived insufficiency to policy support. We are therefore hesitant to propose an overarching framework suggesting that individuals support policy because they are willing to cooperate only if others do. As we point out in the discussion section (p. 30), it is conceivable that individuals support policy for a number of reasons, such as advancing climate change mitigation in recognition of the cumulated effects of individual action, punishing uncooperative others, or, as you point out, to restore social fairness. We have added this last point to our discussion:

→ Given the tendency for individuals to engage in costly punishment when they perceive insufficient cooperation (34,76), potential mediators of the present results include the desire to punish uncooperative others versus alignment with the collective goal (in this case, climate change mitigation). Building on Ostrom’s work on social dilemmas (77), the observed results could also be interpreted as contingent cooperation, such that individuals will cooperate on the condition that others do. Policy governs all citizens and may therefore constitute a means of enforcing cooperation. Future experimental work might compare the effects of the proposed candidate mechanisms (i.e., goal advancement, social punishment, contingent cooperation). (p. 30)

77. Ostrom E. Governing the commons: The evolution of instutions for collective action. Cambridge, UK: Cambridge University Press; 1990. 

3. On a related note, the authors cite the strength of personal goal importance as a predictor of policy support in recommending policymakers make the environmental impacts of their policies as “visible as possible” (p. 31). I do not think these results justify this recommendation. Conceptually, if personal goal importance is the best predictor of policy support, that does not mean that worries about a policy’s connection to the env goal are the biggest threat to support. Indeed, with the findings about worries about the behavior of others, we can interpret these results as or more intuitively (in my view) as indicating that fear of a collective action failure is the biggest threat to policy support at the margin. In other words, even if I fully understand the environmental impacts of the policy, I may not support it if I don’t think it will change behavior among most or all other members of society. This might well be the more important policy implication of the paper, and the authors should consider these possibilities more carefully in writing up their “practical” implications for policymaking.

We absolutely agree that one needs to exercise caution in formulating suggestions for policy design and communication. Our previous suggestion was not yet well-aligned with our findings, and we have therefore reformulated it as follows:

→ Policymakers might, therefore, consider highlighting the importance of proposed policies for climate protection to link to pre-existing personal climate change mitigation goals. (p. 31)

Indeed, fear of a collective action failure may well play an important role in informing individuals’ support for climate policy. However, given that those who perceive others not to be doing enough to protect the climate (i.e., those who may be afraid that not enough is being done) are more supportive of the proposed policy. This suggests to us that the perception of insufficient collective action may drive rather than threaten policy support. In our view, this perspective links into speculation regarding candidate mechanisms linking perceived insufficiency to policy support, as detailed in our response to your second comment. We hope that we have correctly understood and represented your respective and are happy to re-adjust relevant sections if we have missed something.

Reviewer #2: 

This is one of the best manuscripts that I've seen in several decades. The literature review is excellent. The integration of key social psychological theory and findings with the specific goals of this pair of studies is creative and right on target. It is exactly the kind of research that I've been urging social psychology students and colleagues to do for over a decade. Here are a few suggestions for minor changes

Thank you very much for your extremely generous assessment of our manuscript. We are grateful for your encouragement and your suggestions for further improvements, which we address below.

1. Line 271. Include one sample item of the Personal Goal Importance scale in the text.

We do include a sample item, as suggested:

→ To assess participants' personal importance of climate change mitigation, we asked them to consider "the goal of doing something against climate change in your everyday life." Participants indicated the extent of their agreement with each of five items on a scale ranging from 1 (do not agree at all) to 5 (completely agree), e.g., "This goal is important to me" (adapted to the environmental context from (11)). (pp. 11-12)

2. Did the authors consider the possibility of moderation effects? For example, is it possible that trait self-control interacts with perceived insufficiency of others' behavior? Exploratory analyses of this type might prove interesting.

Thank you for your suggestion. It is indeed interesting to consider whether those who are habitually good at employing self-control (i.e., trait self-control) or those who are good at resisting temptation to act more pro-environmentally (not assessed in the current study) would expect others to do the same and might therefore be more strongly in favor of a policy when they see others’ behavior as insufficient. However, the interaction between trait self-control and perceived insufficiency is non-significant, βStudy 1 = -.05, 95% CI [-.17, .08], p = .455, βStudy 2 = -.03, 95% CI [-.12, .06], p = .500. It is also plausible that those to whom climate change mitigation is a more important personal goal more strongly support policy when they perceive others’ actions as insufficient. However, this interaction term is also non-significant, βStudy 1 = -.07, 95% CI [-.16, .03], p = .192, β = -.04, 95% CI [-.12, .04], p = .354. Given that moderation effects require many times the sample size that is required to detect existing main effects (Gelman, 2018) and the present studies are therefore likely underpowered to detect them, we refrain from discussing these analyses in the paper but point out the potential for follow-up work testing moderations in the general discussion.

Gelman, A. (2018). You need 16 times the sample size to estimate an interaction than to estimate a main effect. Statistical Modeling, Causal Inference, and Social Science. https://statmodeling.stat.columbia.edu/2018/03/15/need-16-times-sample-size-estimate-interaction-estimate-main-effect

→ Future studies may also consider the extent to which individuals are aware that punishment of the collective entails regulation of their own behavior, given that it is applied to all members of society, or whether the effect differs in magnitude according to individuals’ own self-control success or goal importance. Given that vastly larger sample sizes are required to detect interaction effects (78), we call for future sufficiently powered studies to test potential moderators. (p. 30)

3.p. 30. Another potentially useful follow-up study would be to replicate this study across a wide range of countries/cultures.

We absolutely agree and have inserted a call for cross-cultural replication in the general discussion:

→ It would be insightful to investigate generalization across other collective action problems and across cultures, and to establish the boundary conditions under which the effect emerges. (p. 29)

4. One of the major problems in the climate change/psychology domain is the relative isolation of scholars in rather narrow towers of specialization. I first became aware of this more that a decade ago when key editors of psychology works were largely uninterested in publishing works that integrated both the geological/physical aspects of climate change with the role played by psychological factors. I offer two specific citations for the authors' consideration, one of which could be seen as quite self serving. This first is a new monograph (released this week) that I co-authored:

We fully agree – especially in research on a complex and extremely pressing societal issue like climate change mitigation, it is vital to integrate findings and perspectives across specializations and, indeed, across disciplines. The first author is currently undertaking work connecting social and physical environments to support for a range of environmental policies and the perceived feasibility of behavior change. Thank you for providing links to your own and work fostering such connections.

Miles‐Novelo, A. and Anderson, C. A. (2022). Climate Change and Human Behavior: Impacts of a Rapidly Changing Climate on Human Aggression and Violence. Cambridge, UK: Cambridge University Press. https://www.cambridge.org/core/elements/abs/climate-change-and-human-behavior/F64471FA47B8A6F5524E7DDDDE571D57

This monograph is free to download from the above link, until Feb. 21. It also can be downloaded from my web site: www.CraigAnderson.org

The second one is a recent article:

Barlett, C. P., DeWitt, C. C., Madison, C. S., Heath, J. B., Maronna, B., & Kirkpatrick, S. M. (2020). Hot temperatures and even hotter tempers: Sociological mediators in the relationship between global climate change and homicide. Psychology of Violence, 10(1), 1-7. http://dx.doi.org/10.1037/vio0000235

Please note that I am urging the authors to add either reference to this paper. The present version is excellent with or without describing either of these.

Craig

Craig A. Anderson, Ph.D.

Editor, Aggressive Behavior

Distinguished Professor of Psychology

Iowa State University

caa@iastate.edu

---

## [Editor Report · Decision Letter 1]

13 May 2022

Climate policy support as a tool to control others’ (but not own) environmental behavior?

PONE-D-21-40332R1

Dear Dr. Kukowski,

I have read your cover letter and the revision carefully, and am happy with the changes you have made. I appreciate your very responsive and thorough revisions, and I am glad to accept your paper for publication.

Therefore, we’re pleased to inform you that your manuscript has been judged scientifically suitable for publication and will be formally accepted for publication once it meets all outstanding technical requirements.

Kind regards,

Cristina Zogmaister, Ph.D.

Academic Editor

PLOS ONE
---

## [Editor Report · Acceptance letter]

26 May 2022

PONE-D-21-40332R1 

Climate policy support as a tool to control others’ (but not own) environmental behavior? 

Dear Dr. Kukowski:

I'm pleased to inform you that your manuscript has been deemed suitable for publication in PLOS ONE. Congratulations! Your manuscript is now with our production department. 

Kind regards, 

on behalf of

Dr. Cristina Zogmaister 

Academic Editor

PLOS ONE